# The Role of Propolis as a Natural Product with Potential Gastric Cancer Treatment Properties: A Systematic Review

**DOI:** 10.3390/foods12020415

**Published:** 2023-01-16

**Authors:** Nelly Rivera-Yañez, Porfirio Alonso Ruiz-Hurtado, Claudia Rebeca Rivera-Yañez, Ivonne Maciel Arciniega-Martínez, Mariazell Yepez-Ortega, Belén Mendoza-Arroyo, Xóchitl Abril Rebollar-Ruíz, Adolfo René Méndez-Cruz, Aldo Arturo Reséndiz-Albor, Oscar Nieto-Yañez

**Affiliations:** 1Carrera de Médico Cirujano, Facultad de Estudios Superiores Iztacala, Universidad Nacional Autónoma de México, Tlalnepantla 54090, Mexico; 2División de Investigación y Posgrado, Facultad de Estudios Superiores Iztacala, Universidad Nacional Autónoma de México, Tlalnepantla 54090, Mexico; 3Laboratorio de Toxicología de Productos Naturales, Departamento de Farmacia, Instituto Politécnico Nacional, Escuela Nacional de Ciencias Biológicas, Av. Wilfrido Massieu, Esq. Manuel L. Stampa s/n, Gustavo A. Madero, Ciudad de México 07738, Mexico; 4Laboratorio de Toxicología Molecular y Celular, Departamento de Farmacia, Instituto Politécnico Nacional, Escuela Nacional de Ciencias Biológicas, Av. Wilfrido Massieu, Esq. Manuel L. Stampa s/n, Gustavo A. Madero, Ciudad de México 07738, Mexico; 5Laboratorio de Inmunología, Unidad de Morfofisiología y Función, Facultad de Estudios Superiores Iztacala, Universidad Nacional Autónoma de México, Tlalnepantla 54090, Mexico; 6Laboratorio de Inmunonutrición, Sección de Estudios de Posgrado e Investigación, Escuela Superior de Medicina del Instituto Politécnico Nacional, Plan de San Luis esq. Salvador Díaz Mirón s/n, Ciudad de México 11340, Mexico; 7Laboratorio de Inmunidad de Mucosas, Sección de Estudios de Posgrado e Investigación, Escuela Superior de Medicina del Instituto Politécnico Nacional, Plan de San Luis esq. Salvador Díaz Mirón s/n, Ciudad de México 11340, Mexico

**Keywords:** propolis, gastric cancer, in vivo and in vitro models, KATO III, cytotoxic effect, bioactive compounds

## Abstract

Gastric cancer is one of the most common, aggressive, and invasive types of malignant neoplasia. It ranks fifth for incidence and fourth for prevalence worldwide. Products of natural origin, such as propolis, have been assessed for use as new complementary therapies to combat cancer. Propolis is a bee product with antiproliferative and anticancer properties. The concentrations and types of secondary metabolites contained in propolis mainly vary according to the geographical region, the season of the year, and the species of bees that make it. The present study is a systematic review of the main articles related to the effects of propolis against gastric cancer published between 2011 and 2021 in the PubMed and Science Direct databases. Of 1305 articles published, only eight studies were selected; among their principal characteristics was the use of in vitro analysis with cell lines from gastric adenocarcinoma and in vivo murine models of the application of propolis treatments. These studies suggest that propolis arrests the cell cycle and inhibits proliferation, prevents the release of oxidizing agents, and promotes apoptosis. In vivo assays showed that propolis decreased the number of tumors by regulating the cell cycle and the expression of proteins related to apoptosis.

## 1. Introduction

### 1.1. Gastric Cancer

Stomach cancer is the fifth most common type of malignant dysplasia worldwide, affecting 18,094,716 people and causing 9,894,402 deaths around the world in 2020. Asian countries including China, Japan and India are among those with the highest incidence and mortality rates, but other nations with predominantly Caucasian populations, such as the United States of America, Germany, and the Russian Federation, were also rated among the ten most affected countries in terms of new cases and deaths in the last year [1,2]. However, incidence and death rates have been decreasing over the last 30 years, and many researchers suggest that this is, at least partly, because of the control and prevention of risk factors and the promotion of healthy habits [3].

Over time, there have been proposals to classify gastric cancer, including those by Lauren, Mulligan, Nakamura, Ming, and Goseki, but nowadays, two of the most popular methods are anatomical division (cardias and non-cardias) and the classification by the World Health Organization (establishing subtypes of adenocarcinoma, signet ring-cell carcinoma, and undifferentiated carcinoma) [4,5,6,7,8,9].

Recently, many studies have suggested that the gastric cancer location and type are related more strongly to some risk factors than others. For example, non-cardias gastric cancer’s principal cause is infection by *Helicobacter pylori*, but aside from this, the consumption of alcohol, salty foods, and grilled barbecued meat and fish, tobacco smoking, and low fiber (fruits and vegetables) intake are also related to this cancer. In addition, there are occupations with high risk, such as those of fishermen, nurses, and machine operators exposed to nitrogen oxides and radiation. Cardias gastric cancer is also related to *H. pylori*, but it is more common in patients aged under 50 years old. Risk factors include prolonged use of non-steroidal anti-inflammatory drugs (NSAIDs), obesity, Epstein–Barr virus infection, and post-surgical gastric remnants. Cardias and non-cardias cancer are both related to blood group A, having a low socioeconomic position, poor hygiene measures, and male gender [10,11,12]. On the other hand, it is well established that the most effective prevention measures are the consumption of a healthy diet (Mediterranean and low-sodium diets) and the consumption of fresh fruit and dark green, light green, and yellow vegetables, which contain beta carotene, vitamin C, vitamin E, and folate [13,14]. Recently, many researchers have focused on finding therapeutics other than chemotherapy and surgery to treat cancer.

In this context, the administration of distinct natural products, including propolis and its different components that confer anti-inflammatory, antioxidant, and antiseptic properties to propolis, has been proposed [15]. This review concentrates on the antiproliferative, cytotoxic, and anticancer activity of this natural bee-derived product [16].

### 1.2. Propolis

Many products of natural origin have the ability to produce metabolites that can generate benefits for humans, but in many cases, these compounds can only be extracted from plant sources. However, bees are architects of the production of different bee products using plant resources from their environments, one of which is propolis [17]. Propolis is a natural resinous product produced by bees to build and defend their hives. In fact, the word propolis comes from the Greek words “pro” (defense) and “polis” (city), meaning “defense of the city” or “defense of the hive” [18,19,20]. Throughout history, different cultures have made use of propolis; for example, Egyptians used propolis to mummify their deceased, the Greeks and Romans used it as a topical ointment for the treatment of wounds, cuts, and ulcers, and in Eastern Europe, its use was recurrent, earning it the nickname “Russian penicillin” [18].

It is important to mention that propolis tends to be very diverse in terms of its chemical composition, since it is affected by botanical origin, the edaphoclimatic conditions, the season of the year, and the bee species [21,22]. Significant differences in propolis can be found within a country in terms of the composition and biological properties of propolis from different climatic regions. This is the case for the different types of propolis found in Brazil, which have been classified by color and texture. Due to the above reasons, it is not possible to mention unique characteristics for propolis [19]. As previously mentioned, the composition of propolis is extremely complex, since bees look for different sources to make it, such as resins, flower excretions, leaves, buds, shoots, stems, and fruits. This mixture of raw materials results in more than 600 different chemical compounds that make up propolis [23,24]. Different investigations have elucidated various chemical groups, among which esters, flavonoids, terpenes, aldehydes, and aromatic alcohols stand out, as well as fatty acids, stilbenes, and steroids [23,25]. However, recent research has mentioned that flavonoids, phenolic acids, and terpenoids are considered the main biologically active substances in propolis [17,26].

For years, the search for biologically active substances to use against various diseases has focused mainly on natural sources, one of them being propolis. Different investigations have shown the potential of propolis to act against different types of cancer, including brain, head and neck, tongue, breast, liver, colon, cervix, prostate, pancreas, kidney, bladder, blood, and skin cancer [27,28,29,30,31,32,33,34,35]. In addition, it has been reported that different compounds identified in propolis can exert their activity through various genetic and biochemical pathways of cancer progression, although these effects vary depending on its botanical origin, geographic region, and method of extraction and subsequent preparation [28]. The foregoing information positions propolis as a viable alternative and a source of different compounds that may contribute to cancer therapy.

## 2. Materials and Methods

This systematic review was conducted following the criteria of Preferred Reporting Items for Systematic Reviews and Meta-Analyses (PRISMA); the search process was carried out by employing the PubMed (https://pubmed.ncbi.nlm.nih.gov (accessed on 26 January 2022)) and Science Direct (https://www.sciencedirect.com (accessed on 26 January 2022)) databases based on the research processes used by other authors [36,37,38]. Only research articles published between 2011 and 2021 were included in this work, and the terms used to search both databases were “propolis and gastric cancer, propolis and stomach cancer, propolis and stomach carcinoma, propolis and gastric carcinoma, propolis and KATO-III, propolis and gastric cancer cell line”.

The selection criteria included articles published in the English language with the following characteristics: (1) original and full-text papers; (2) research works that evaluated gastric cancer models or cell lines related to gastric cancer; and (3) papers that used propolis as a treatment model. Exclusion criteria included the following points: (1) articles written in another language; (2) review articles; (3) papers from news, editorial letters, or social media; and 4) duplicated studies. All search procedures, including the paper selection process, are summarized in Figure 1.

The figures and tables were designed by the authors of this review with Microsoft PowerPoint (16.43) software and the scalable vector graphics editor Inkscape (1.0.2) and were edited and escalated with the GNU image manipulation program GIMP (2.10.22). Therefore, all figures and images do not have copyright issues.

## 3. Results

### 3.1. Selected Papers and Characteristics of Studies

From the selection process shown in Figure 1, a total of 1305 papers related to the terms used in both databases were identified. Most articles were located in the Science Direct database (1266 records), whereas only 39 articles were recorded from the PubMed database. Of these, only eight papers were selected for inclusion, because they met both the selection and exclusion criteria named in the material and methods section.

The selected articles were published in seven indexed journals, which are listed below with their journal impact factor (JIF) according to InCities Journal Citation Reports (https://jcr.clarivate.com/jcr/home accessed on 19 March 2022): BMC complementary and alternative medicine with a JIF of 3.659 (renamed BMC Complementary Medicine and Therapies without a JIF) [39,40]; Asian Pacific Journal of Tropical Biomedicine with a JIF of 1.545 [41]; Asian Pacific Journal of Cancer Prevention with a JIF of 2.514 until 2014 (currently this journal does not have a JIF) [42]; Journal of Functional Foods with a JIF of 4.451 [43]; Evidence-Based Complementary and Alternative Medicine with a JIF of 2.630 [44]; Archives of Iranian medicine with a JIF of 1.354 [45]; and Scientific Reports with a JIF of 4.380 [46].

The studies included in this review and their general characteristics are summarized in Table 1. The propolis samples evaluated by the different research groups had many origins, including different types of bees. In this regard, one study worked with propolis from *Apis mellifera* bees [40]; four studies worked with various types of propolis from stingless bees, including *Trigona laeviceps* [39], *Trigona incisa* [41,42], *Trigona apicalis*, *Trigona fuscobalteata*, *Trigona fuscibisca* [41], and *Melipona* (obtained from the UPLB Bee Program Meliponary, University of the Philippines) [46], and three studies did not specify the type of bee that produced the propolis with which they worked [43,44,45].

It should be noted that, although the propolis samples were used as ethanolic and methanolic extracts, there were differences in the methods employed to obtain these extracts. In works that used ethanol as a solvent [43,44,45,46], the origin of propolis samples varied depending on the authors. Catchpole et al. [43] used propolis samples obtained from commercial manufacturers of New Zealand propolis tinctures (Manuka Health) with 25% or 40% dissolved solids. The manufacturer reported that these tinctures contain at least 300 mg/g of different secondary metabolites (phenolic compounds) on a wax-free basis, and the used propolis was dissolved in a mixture of ethanol/water solvent. These propolis samples were encapsulated in alpha, beta, and gamma cyclodextrins (α-CD, β-CD, and γ-CD respectively), and propolis-cyclodextrin complexes were obtained as the final products. The use of commercial sources of propolis has the advantage of a consistent propolis composition that is not dependent on the region from which the raw propolis used by the manufacturers was collected.

In another case, a Chinese propolis sample collected from the Changbai Mountain area was twice sonicated with 95% ethanol in an ultrasonic water bath to obtain an ethanolic propolis extract. With this, the authors formulated two solutions with final concentrations of 50 mg/mL and 10 mg/mL, respectively [44]. Similarly, propolis samples were collected from the Hamadan and Taleghan districts of Iran from areas with the presence of Poplar and *Ferula ovina* plants in the fall of 2010. Ethanolic extract was obtained by maceration with 96% ethanol and then dissolved in dimethyl-sulfoxide (DMSO) to obtain a final concentration of approximately 500 mg/mL [45]. Standardized and authenticated Philippine stingless bee propolis was used by Desamero et al. [46]. They obtained an ethanolic extract with a final concentration of 300 mg/mL after the maceration process with the use of analytical grade ethanol as the solvent.

In the same context, other groups of researchers have worked on the extraction of different types of propolis with ethanol or methanol to obtain different partitions and fractions using a variety of column chromatographies and solvents [39,40,41,42]. One case was a sample of stingless bee propolis from *T. laeviceps*, which was collected in central Thailand from an apiary in Samut Songkram province. The propolis extraction process was carried out twice with 95% ethanol in agitation. It was centrifuged to clarify it and obtain the propolis ethanolic extract. In the same study, the authors partially purified the ethanolic extract of propolis with solvents of different polarities (hexane, dichloromethane, and methanol) to obtain various extracts. Similarly, different partitions were obtained from the hexanic extract by quick column chromatography of silica gel, and finally, from these last partitions obtained by columns with 30% and 100% dichloromethane, different fractions were obtained by size exclusion chromatography using a Sephadex LH-20 column [39].

In another investigation, propolis was collected from a Thai apiary in Nan province in the Pua district between January and February 2010. The authors twice extracted propolis with 80% methanol, stirred it, and clarified it to obtain the methanolic extract of propolis, from which they subsequently obtained an extract with dichloromethane and then obtained a hexane extract. From the hexanic extract, the authors obtained various fractions by quick column chromatography with silica gel [40].

Several Indonesian propolis samples were obtained by Kustiawan et al. [41] from Mulawarman University Botanical Garden, Samarinda, East Kalimantan from different stingless bees, such as *T. fuscibisca*, *T. fuscobalteata*, *T. apicalis*, and *T. incisa*, in February 2013. They were extracted several times with 96% methanol while stirring until a light tone coloration was observed in each propolis sample. This was done for a maximum of 7 days to obtain the initial methanolic extract from each propolis sample, and from these samples, the hexanic extracts of propolis were obtained. Later, ethyl acetate extracts of propolis were obtained, and finally, the remainder of each of the samples that did not dissolve after processing with these solvents was used to produce methanolic extracts of propolis.

Similarly, other authors collected propolis from the Indonesian stingless bee *T. incisa* from the same place and on the same date as Kustiawan et al. [41]. They extracted the propolis three times with 96% methanol until a light color was observed to obtain the initial methanolic extract of propolis, and from this sample, they obtained propolis extracts of methanol, hexane and ethyl acetate in the same way as described above by Kustiawan et al. [41]. Subsequently, the authors obtained various fractions from the ethyl acetate extract of propolis by quick column chromatography with silica gel. From the fractions that showed cytotoxic activity, other fractions were obtained by absorption column chromatography with silica gel, and finally, from the fractions that showed activity on KATO-III cells, other fractions were obtained, again by size exclusion column chromatography [42].

It should be noted that, in several cases, the chemical composition of the propolis extract was determined by chromatographic assays. One study analyzed propolis from Thailand using NMR and ESI-MS [40]; another study analyzed propolis from Indonesia by one-dimensional thin layer chromatography (1D-TLC), NMR, and ESI-MS [42]. Another study analyzed propolis from China only using HPLC [44]; one more article analyzed Iranian propolis by means of ultra-performance liquid chromatography-mass spectrometry (UPLC-MS) [45]; and finally, an article analyzed propolis from the Philippines using gas chromatography/tandem mass spectrometry (GC/MS/MS) [46]. In three cases, the chemical composition of the propolis extract was not determined [39,41,43], although in one case, the secondary metabolites reported in New Zealand propolis were given by the manufacturer of this sample [43], as shown in Table 1.

Most works only evaluated the activity of propolis at the in vitro level (Figure 2) [39,40,41,42,43,44], and only two studies evaluated the effect of propolis in rat [45] or mouse [46] cancer gastric models (Figure 3). Of these, one study aimed to evaluate the properties of propolis only at the in vivo level [45]. In contrast, another study evaluated the effects of propolis at both the in vitro and in vivo levels [46].

In terms of studies with an in vitro approach, although a variety of gastric cancer cell lines have been used to determine the anti-gastric cancer effect of propolis (Figure 2), the authors of four works of the eight reviewed used the gastric carcinoma KATO-III cell line to evaluate the activity of propolis from Thailand or Indonesia [39,40,41,42]. We also found three other articles that used a gastric cancer cell line other than KATO-III as an in vitro model. Among these were the human gastric cancer cell lines NCI-N87 [43] and SGC-7901 [44], which were used to evaluate the activity of propolis from New Zealand and China, respectively. One article evaluated a battery of gastric cancer cell lines, including AGS, MKN-45, NUGC-4, and MKN-74 [46], to determine the effect of propolis from the Philippines. In four of the seven in vitro studies, cell lines from the liver (CH-liver) and fibroblasts (HS-27) [39], non-transformed human foreskin fibroblasts (Hs27) [40], normal skin fibroblasts (CCD-986sk) [42], or human embryonic kidneys (HEK293) [44] were implemented as comparative controls. As a positive control, the authors used 5-fluorouracil (5-FU), a cytotoxic and antineoplastic drug [43], or Cisplatin (a cytotoxic drug) [46] in two of the seven in vitro studies.

Concerning works that included in vivo assays, there were differences in the form of gastric cancer induction used in in vivo models (Figure 3). Some authors [45] used MNNG as the reagent to induce tumors in gastric tissue; this was administered to Wistar rats in drinking water, which was provided ad libitum, at a concentration of 100 μg/mL for 34 weeks. It was administered by 10% sodium chloride in drinking water given weekly to animals in the first 6 weeks of life to enhance the development of gastric cancer. Desamero et al. [46] used A4gnt KO mice, which display full-blown differentiated-type gastric adenocarcinoma, because the A4gnt gene (that encoding for α1 glycosyltransferase) deletion induces the complete loss of αGlcNAc (α1, 4-linked N-acetylglucosamine residues) expression, producing the progressive development of differentiated-type gastric adenocarcinoma confined in the pyloric mucosa of mutant mice. C57BL/6J mice, as wild-type animals that did not present any process of carcinogenesis, were used as the control group.

Regarding the parameters evaluated, in vitro studies included cytotoxic assays using the MTT test [39,40,41,42,43,46] and CCK8 test [44] to determine the anti-gastric cancer activity of propolis. In one of the in vitro studies [39], concentrations of 3.125, 6.25, 12.5, 25, 50, and 100 μg/mL were used, and another study [40] used Thailand propolis concentrations of 0.1, 1, 10, 100, and 1000 μg/mL to determine the cytotoxic activity on KATO-III cells by MTT. Other authors [41] tested a single concentration of 20 μg/mL, and another study [42] utilized serial concentrations of less than 10 μg/mL of Indonesian propolis (unspecified) to evaluate the activity in KATO-III cells by MTT.

Catchpole et al. [43] evaluated the effects of a single concentration (200 µg/mL) of pinocembrin and different cyclodextrin complexes of New Zealand propolis against the gastric cancer cell line NCI-N87 to determine their cytotoxic activity by MTT. Desamero et al. [46] worked with four human gastric cancer cell lines, AGS, MKN-45, NUGC-4, and MKN-74, to evaluate the anti-gastric cancer activity of Philippine propolis ethanolic extract at concentrations of 1000, 500, 250, 100, 10, and 1 μg/mL by the MTT test at 24 h, 48 h, and 72 h, respectively. Moreover, the authors determined cell cycle arrest through a flow cytometry assay of the IC_50_ of the propolis extract at 48 h (equivalent to 188 μg/mL). With this same concentration, through DNA fragmentation by employing the TUNEL assay, they measured the potential of propolis to induce apoptosis and used qRT-PCR analysis to investigate propolis-induced modifications in the gene expression of ACTB, BCL2, BCL2L1, BAX, BAD, TP53, CDKN1A, CDKN1B, CCND1, CDK1, and CDK2.

Jiang et al. [44] used the human gastric cancer cell line SGC-7901 to evaluate the cytotoxic activity of their propolis extract using the CCK8 test with propolis concentrations of 6.25, 12.5, 25, 50 and 75 μg/mL. They also used the microscopical test to determine the morphological changes in SGC-7901 cells and their interactions with the concentrations of propolis. In a similar form to Desamero et al. [46], these authors evaluated cell cycle arrest with flow cytometry, although this was not the unique parameter that they evaluated with this technique. They also determined apoptosis induction with propolis extract, changes in the cell membrane permeability, and the generation of ROS. Finally, they used the Western blot assay to determine changes in the expression of several genes (see Table 1).

Similarly, the in vitro cytotoxic activity of different fractions obtained from propolis from Thailand was tested in two investigations. Concentrations of 3.125, 6.25, 12.5, 25, 50, and 100 μg/mL were tested in one article [39], and the other study utilized concentrations of 1.5, 3.125, 6.25, 12.5, 25, 50, 100, and 200 μg/mL [40]. Finally, another article used serial concentrations (unspecified) of less than 10 µg/mL of each fraction obtained from propolis from Indonesia, all against KATO-III cells [42].

Concerning the in vivo studies, although the model of gastric cancer induction differed between works, there were some similarities regarding the parameters evaluated, at least at the macroscopic and histological levels. In both studies, the authors examined the alterations presented in the gastric mucosa after treatment with propolis extracts (Iranian propolis or Philippines propolis). Nevertheless, in the case of Iranian propolis (500 mg/mL added to the food, begun 2 weeks before MNNG administration), the characterization of damage at the macroscopical level was more detailed in terms of describing the number, size, and localization of tumors present in rats. Moreover, the authors conducted laparotomy and autopsy studies in mice that died before the end of the experimental procedure [42]. In comparison with propolis from the Philippines (100 mg/kg for 30 days via feeding tube), the authors only described the mucosal stage in the experimental groups and the changes between the experimental groups that they evaluated [46].

On the other hand, both studies utilized a semi-quantitative scoring system to describe the alterations observed at the histological level. In these works, H&E staining was performed to describe the microarchitecture of the gastric cancer tissues in the experimental groups. In the case of Iranian propolis, histological evaluations included the measurement of four parameters: the nuclear/cytoplasmic ratio, epithelial stratification, nuclear dispolarity, and structural abnormalities. Additionally, the histological exams were conducted by two independent researchers and confirmed by a pathologist to obtain consensus on the evaluations [45]. For Philippine propolis, histological measurements were focused on the changes in gross gastric mucosa elevation and the mucosal thickness of the pyloric mucosa. This was because the type of gastric cancer used by the authors was a differentiated-type gastric adenocarcinoma that is usually localized in the pylorus portion of the stomach, unlike diffuse-type gastric cancer, which can be found in any portion of the stomach [46].

Immunohistochemistry studies were used in both works. In one study, Wistar rats were treated with Iranian propolis and changes in the presence of β-catenin, Bax, and Bcl2 antibodies were evaluated by semi-quantitative measurement of the levels of these antibodies in the experimental groups [45]. In contrast, in A4gnt KO mice and C57BL/6J mice treated with Philippine propolis, the levels of CD3 (a marker of T cell expression, BrdU (a proliferation marker), and p-21 (a marker of the cell cycle) were quantified. Moreover, in this last study, the authors also included the evaluation of the gene expression of Actb, Il10, Il11, Il1b, Tnfa, Ifng, Il6, Bcl2, Bcl2l1, Bax, Bad, Trp53, Cdkn1a, Cdkn1b, Ccnd1, Cdk1, and Cdk2 in homogenates of the stomach obtained from the different experimental groups that they evaluated. This analysis was performed through qRT-PCR [46].

### 3.2. Benefits of Propolis for Gastric Cancer in Cell and Animal Models

The benefits of propolis for gastric cancer have been described in various investigations (Table 1) [39,40,41,42,43,44,45,46]. Ethanolic extract of propolis from Thailand showed cytotoxic activity on the KATO-III cell line, with an IC_50_ of 22.98 μg/mL. In contrast, the hexane extract, obtained from ethanolic extract, showed an IC_50_ of <20 μg/mL. In addition, the partitions obtained from the hexane extract with 10%, 30%, 50%, 70%, or 100% dichloromethane, and another with 10% hexane, also showed an IC_50_ of <20 μg/mL. The fractions obtained from the partitions with 30% and 100% dichloromethane presented higher activity, with IC_50_ values of 18.07 μg/mL and 4.09 μg/mL for F2-30% and F3-30%, respectively, and IC_50_ values of 7.55 μg/mL and 8.31 μg/mL for F3-100% and F4-100%, respectively [39].

Similarly, another study conducted with propolis from Thailand reported that dichloromethane and hexane extracts had cytotoxic effects on KATO-III cells with IC_50_ values of 42.5 ± 6.61 μg/mL and 43.8 ± 6.5 μg/mL, respectively. In addition, different fractions obtained from the hexanic extract of propolis were also shown to have activity on this cell line, such as fractions III, IV, and V with IC_50_ values of 13.69 ± 1.44 μg/mL, 40.16 ± 2.66 μg/mL, and 15.21 ± 2.13 μg/ mL, respectively. Subsequently, using NMR and mass spectroscopy, two compounds that also showed cytotoxic activity were identified, a cardanol in fraction III with an IC_50_ of 13.71 ± 1.42 μg/mL and a cardol in fraction V with an IC_50_ of 8.78 ± 0.28 μg/mL. However, although the authors were able to identify two compounds in Thai propolis, they suggested that there are other bioactive components and there could even be a synergism between them to present this cytotoxic activity [40].

One more investigation showed that, out of a total of 12 Indonesian propolis extracts obtained from four different stingless bee species and with three distinct solvents, only seven had cytotoxic activity against KATO-III cells. In this context, all of the extracts were tested at a concentration of 20 μg/mL, and it was observed that the methanolic extracts of propolis from the bee species *T. incisa*, *T. fuscobalteata*, and *T. fuscibasis* inhibited cell proliferation by 82% ± 0.001, 80% ± 0.008, and 55% ± 0.040, respectively. Subsequently, ethyl acetate and hexane extracts were obtained from the methanolic extract of propolis, and it was found that ethyl acetate extracts of propolis from *T. incisa*, *T. fuscobalteata*, and *T. fuscibasis* decreased cell viability by 83% ± 0.001, 82% ± 0.003 and 60% ± 0.016, respectively. In addition, hexanic extract of propolis from *T. apicalis* inhibited this cell line by 57% ± 0.011. These authors did not perform an analysis of the chemical compositions of the different propolis samples from Indonesia to identify their components; however, they indicated that compounds previously reported in propolis and tested in this study at a concentration of 10 μg/mL, such as kaempferol and apigenin, presented cytotoxic activity and decreased the viability of KATO-III cells by around 65% and 55%, respectively [41].

Kustiawan and collaborators [42] also worked with propolis from Indonesia, testing the cytotoxic activity of three different extracts obtained from the propolis of the bee species *T. incisa* on KATO-III cells, as well as different fractions obtained from the methanolic extract of propolis. In this regard, methanolic extract of propolis from Indonesia showed activity with an IC_50_ of 6.06 ± 0.39 μg/mL. Subsequently, ethyl acetate and hexane extracts were obtained from the methanolic extract of propolis, but only the ethyl acetate extract had activity on this cell line with an IC_50_ of 8.06 ± 0.08 μg/mL. In addition, of the various fractions obtained from the methanolic extract of propolis, only fractions F24, F26, F27, F36, F45, and F46 presented cytotoxic activity with IC_50_ values of 9.35 ± 0 μg/mL, 6.75 ± 1.15 μg/mL, 6.61 ± 1.29 μg/mL, 9.75 ± 0.19 μg/mL, 6.06 ± 0.39 μg/mL, and 8.25 ± 0.22 μg/mL, respectively. Finally, through NMR and ESI-MS analyses, cardol was identified in F45, and a terpenoid-like pattern was observed in F46.

In another study, in addition to the in vitro cytotoxic activity of New Zealand propolis (provided by the company Manuka Health), the antioxidant and anti-inflammatory activity was also investigated by authors who formulated different CD complexes containing propolis, encapsulated them, and tested the efficacy of each one. They produced the encapsulates by mixing different concentrations of propolis and distinct CD complexes. For CD1 and CD2, they used 25% solids of propolis and γ-CD, but CAPE was also added to CD2 to double the concentration of this compound; for CD3, they implemented 40% solids of propolis and γ-CD; and for CD4 and CD5, they mixed 40% propolis tincture and α-CD or β-CD. It was found by HPLC analysis that these five New Zealand propolis encapsulates contained compounds such as CAPE, pinobanksin, pinobanksin-3-acetate, pinocembrin, chrysin, and galangin at various concentrations. To evaluate the anti-inflammatory activity possessed by propolis and two of its previously produced complexes, they determined the percentage of TNF-α inhibition in rat blood neutrophils stimulated with LPS, reporting that the sample with New Zealand propolis alone and those with CD1 and CD2 inhibited TNF-α by 85% ± 1, 93% ± 1 and 97% ± 1, respectively, at a concentration of 50 μg/mL. In addition, these three tested samples inhibited this cytokine by 100% at a concentration of 200 μg/mL [43].

Regarding the antioxidant activity, it was observed through the β-carotene bleaching assay, which provides a measure of the antioxidant activity of lipids, that the five propolis complexes and CAPE (also in the γ-CD complex) presented moderate antioxidant activity. The only one that presented strong antioxidant activity was CAPE alone. Finally, these authors investigated the cytotoxic effect of the CD3, CD4 and CD5 complexes and that of pinocembrin on the gastric cancer cell line NCI-N87, reporting that the three propolis complexes had moderate cytotoxic activity, since they only inhibited 32.7%, 24.6% and 21.8% of these cells, respectively, at a concentration of 200 μg/mL. It is worth mentioning that, in this trial, it was observed that pinocembrin inhibited NCI-N87 cells by 72.5% at a concentration of 200 μg/mL. In this research, the authors found that New Zealand propolis had bioactive components with antioxidant, anti-inflammatory, and cytotoxic activity, and it was mentioned that the study of CD and propolis complexes could be a promising topic for future research on this type of cancer [43].

An in vitro study performed by Jiang et al. [44] on the SGC-7901 gastric cancer cell line showed that Chinese propolis reduced the viability of these cells. They also calculated the IC_50_ of this propolis, which was 66.64 μg/mL. They recorded changes in the normal morphology of cancer cells. The treatment with Chinese propolis altered the structure; in other words, gastric cancer cells were reduced in size and density in contrast with cells that were not treated with propolis. A relationship between the degree of morphological alteration and the concentration of propolis used in their experiments was shown. This was related to the capacity of Chinese propolis to induce ROS production in the SGC-7901 gastric cancer cell line. This was explained by the capacity of treatments to induce a reduction in mitochondrial membrane permeability in a dose-dependent manner, which was closely related to the generation of ROS in the inner part of cells and also related to the induction of apoptosis produced by treatment with propolis in cancer cells. The molecular mechanisms that the authors associated with the ability of Chinese propolis to decrease the viability of SGC-7901 cells were induced by the up-regulation of Bax and Bid proteins, the activation of p53, as well as the activation of cleaved caspase 8, cleaved, caspase, 9, cleaved caspase 3, and cleaved PARP. In contrast with the down-regulation of Bcl-2 (an anti-apoptotic protein), it is important to consider that the liberation of cytochrome C was also observed in this study. Additionally, the anticancer mechanisms associated with Chinese propolis have also been shown to have the capacity to induce arrest in the S-phase of the cell cycle in SGC-7901 cells when G1-phase cells were decreased to interact with propolis treatments. This last point is related to the up-regulation of the proteins associated with cell cycle expression that propolis displayed in P-Rb, CDC2, CDK2, cyclin E, cyclin A, and E2F1 in a dose- and time-dependent manner.

Other authors [45] also investigated the in vivo anti-gastric cancer properties of two samples of Iranian propolis. In a model of gastric cancer induced with MNNG, rats developed lesions, dysplasia, and gastric adenocarcinoma with a greater incidence in the stomach than in the intestines or colon. The incidence, number, and sizes of tumors were reduced in rats treated with both samples of Iranian propolis. The level of reduction was from 30% to 40% compared with untreated rats. These observations were confirmed with histological examinations of the experimental groups. As a possible mechanism of action for these Iranian propolis samples, the increase in the level of proapoptotic Bax protein in parallel with the reductions in the levels of expression of β-catenin and antiapoptotic Bcl-2 protein in propolis-treated rats were proposed due to observations of control rats.

Additionally, the capacities of other types of propolis found in different zones around the world to reduces the viability of other gastric cancer cell lines have been evaluated. Desamero et al. [46] analyzed the effects of Philippine propolis from stingless bees on the differentiated-type cancer cell lines AGS, NUGC-4, and MKN-45, which were found to be sensitive to treatment with different concentrations of stingless bee propolis in a dose- and time-dependent manner. The antiproliferative capacity was more remarkable at higher concentrations of propolis and higher treatment incubation times. The diffuse-type gastric cancer cell line MKN-74 was found to be less sensitive to the action of propolis because at higher concentrations and interaction times, it presented IC_50_ values of greater than 900 μg/mL.

To explain the capacity of propolis from the Philippines to inhibit the viability of diverse gastric cancer cell lines, the authors measured the expression of diverse genes related to apoptosis and the cell cycle. Of these processes, through the up-regulation induced by propolis of CDKN1A in AGS and NUGC-4 cells and, although it is unclear, in MKN-45 cells, it was shown that the expression of this gene increases when treated with propolis. Moreover, increases in the levels of CDKN1B and TP53 were observed in AGS cells, while NUGC-4 and MKN-45 cells showed the down-regulation of CDK2. In contrast, MKN-74 cells did not show any changes in the expression of CDKN1A. However, propolis treatment has been shown to decrease the expression of CDK1 and CCND1. This is related to the propolis-induced arrest of the cell cycle in the G0/G1 phase in AGS cells, followed by an increment of cells in the S-phase with the depletion of cells in the G2/M and multinuclear phases. Referring to the expression of genes associated with the apoptosis process, there is not a clear pattern. In AGS cells, propolis treatment induced the expression of Bax and Bad. In contrast, in MKN-45, only down-regulation of Bcl-2L1 was shown, and in NUGC-4 cells, down-regulation of the expression of genes was only observed in Bcl-2. Moreover, a marked increase in DNA fragmentation was shown in MKN-45 cells with slight increases in AGS and NUGC-4 cells. Specifically, MKN-74 apoptotic gene expression was not altered by treatment with stingless bee propolis, which is in accordance with the lower activity that propolis has against this diffuse-type gastric cancer cell line [46].

Interestingly, in this work, the authors also evaluated the in vivo anti-gastric cancer activity of propolis from the Philippines [46]. This was evaluated in A4gnt KO mice that spontaneously developed gastric adenocarcinoma, a type of cancer that is consistent with differentiated-type gastric cancer. As a wildtype, they used C57BL/6J mice. After 30 days of propolis treatment, the stomachs of A4gnt KO mice showed a regression of gross mucosal elevation and a decrease in the thickness of the gastric mucosal layer at the histological level. This contrasted with what was observed in untreated mice. Additionally, in A4gnt KO mice, propolis treatment reduced the infiltration of CD3-positive T-lymphocytic cells. It is worth mentioning that neither treated nor untreated wildtype C57BL/6J mice showed differences regarding the gross morphology, gastric mucosa thickness, or the infiltration of T-lymphocytes. Therefore, the authors highlighted the promissory anti-tumor activity of propolis from the Philippines. In line with this, the overexpression of genes related to the cell cycle seems to be the principal anti-tumor activity of propolis because both treated A4gnt KO mice and wildtype C57BL/6J mice showed increased upregulation of CDKN1A. Moreover, treated A4gnt KO mice also showed increased expression of CDKN1B and reduced expression of CDK1. Additionally, mice treated with propolis had strongly reduced numbers of actively dividing BrdU-positive S-phase cells. In contrast, gene expression related to the apoptosis process did not show a clear result, which could be explained by the lower increase in the DNA fragmentation in treated A4gnt KO mice in contrast to that in untreated mice; therefore, the apoptosis process could have trivial relevance to the anti-gastric cancer activity of propolis.

## 4. Discussion

Propolis is a complex bee product with diverse biological activities that are determined by its variable chemical composition [17,23,36,47,48,49,50,51]. This, in turn, depends on different factors such as the geographical region in which it is collected, the surrounding flora, and the bee species that produces it [50,52,53].

Regarding the bee species, around 600 species of stingless bees have been described [54], with the genera *Melipona*, *Trigona*, *Trigonisca*, *Plebeia* and *Scaptotrigona* having the greatest numbers of known bee species [55]. These species are distributed across a wide tropical geographic region including Central and South America, tropical Africa, southeast Asia, and northern Australia [54]. It has also been described that the *Apis* genus of bees contains around 10 known species, within which *A. mellifera* has a wide native distribution worldwide, including in Asia, Africa, and Europe. It has been spreading due to the activities of beekeepers [56].

Taking the above into account, in the articles included in this review, we found that the activity of propolis from both stingless bee species [39,41,42,46] and *A. mellifera* bees [40] was studied using different models of gastric cancer. Three works did not specify the type of bee employed [43,44,45]; however, we suspect that *A. mellifera* was used in these cases, since it is known that this species is easily handled in hives, making its propolis easier to access [40]. Although these eight investigations did not emphasize or make a correlation between the type of bee that produces the propolis and the effect it has, a few reports have attempted to describe this correlation. Silici et al. [57] mentioned that three samples of propolis collected from the same apiary but from hives of different subspecies of *A. mellifera* presented distinct antibacterial activities, with the propolis from *A. mellifera caucasica* presenting greater activity than that from *A. mellifera carnica* and *A. mellifera anatolica*. However, the authors also mentioned that, based on the chemical composition found for each propolis, the three bee species did not use a single botanical source to produce their propolis. The same was found for propolis produced by stingless bees, as reported in a review by Popova et al. [50]. This is a controversial situation because some works have mentioned that the same species of bee found in distinct regions collects the same resin, while other works could not determine whether there is a relationship between the chemical composition or the origin of the propolis and the bee species. Some research has even found that both *A. mellifera* bees and stingless bees use plants that are available and closest to their hives and not because they have a preference for these plants [58,59]. However, more studies focused on investigating this correlation are needed. Therefore, the species of bee as well as the type of flora in the region are factors that determine the variability of the chemical composition and the activities of propolis [59,60].

Returning to the diversity of botanical sources that bees rely on to produce propolis, a factor involved in the variability of the chemical composition of propolis, only one [45] of the eight works reviewed here mentioned the probable botanical source that the bees utilized for propolis production. However, they did not chemically compare the composition of propolis and the mentioned plants. In that study, the authors [45] stated that Iranian propolis was collected in regions where the *Ferula ovina* and *Poplar* plants are distributed. Similarly to the case of bee species, few investigations of propolis activities have also researched the types of plants that are characteristic of the region or the botanical sources used by bees to produce propolis.

Although it is a great advance that the publications that work with propolis mention the country of origin and the surrounding botanical sources, there is a need to specify and chemically compare the propolis samples with the different plants used by bees to produce propolis. In a review carried out in 2000 by Bankova et al. [52], it was mentioned that the first investigations that emerged in the 1970s chemically compared the flavonoids of propolis with those of poplar in France or with birch in Russia. In addition, Bankova et al. [52] mentioned that a few studies conducted between the 1980s and 1990s demonstrated through chemical analysis that, for countries in temperate zones, such as North America, Europe, Asia, and even New Zealand, *Populus* species are used by bees for the production of propolis. All of this information was also included in a recent review by Dezmirean et al. [61], who also mentioned that a few publications from the 1970s and 1980s chemically described one of the botanical sources used by bees in northern Russia: *Betula verrucosa*. Based on scientific evidence, propolis derived from plants of the *Populus* genus as the main botanical source is known as poplar-type propolis [52]. In his review, Bankova et al. [52] also mentioned that, for tropical areas, in the 1990s, it was chemically verified by a couple of authors that *Cistus* spp. are used to produce propolis in Tunisia, *Ambrosia deltoidea* and *Encelia farinosa* are used for propolis production in the Sonora Desert, and *Clusia minor* and *Clusia major* are used for propolis production in Venezuela.

In the 21st century, and mainly in the last decade, various chemical analyses have been used to show, based on chemical profiles, that the number of botanical sources used by bees to produce propolis in different parts of the world has increased. As reported by different authors, who verified their results by HPLC [62] or HPLC/ESI-MS [63], *Macaranga tanarius* is the plant used by bees to produce propolis in Okinawa, Japan and Hawaii. Other studies have used TLC and NMR to show that *Lepidosperma* sp. Flinders Chase, *Lepidosperma viscidum*, and *Dodonaea humilis* are the botanical sources for propolis produced on Kangaroo Island, Australia [64]. In addition, different studies have verified, using HPLC and NMR, that *Acacia paradoxa* [65] and *Lepidosperma* sp. Montebello [66] are the plants used to produce this same propolis, and these same analyses have been used to show that *Populus fremontii* and *Ambrosia ambrosioides* are the botanical sources of propolis from the state of Sonora in Mexico [67]. Wang et al. [68] determined, by HPLC, that *Populus canadensis* is the botanical source for Chinese propolis from Shandong province, while for propolis produced in the Changbai Mountains in northeastern China, it was verified by HPLC and HPLC/ESI-MS that the *Populus davidiana* dode and *Populus simonii* Carr plants are visited by bees [69]. Another investigation used liquid chromatography with diode array detection coupled with electrospray ionization tandem mass spectrometry (LC/DAD/ESI-MS) and HPLC to show that *P. canadensis* and *Cistus ladanifer* are the botanical sources of propolis from Portugal [70]. Bertrams et al. [71] also verified, by thin-layer chromatography/mass spectrometry (TLC/MS), that for propolis from southern and central Germany, the plants used are *Populus tremula* and *P. canadensis*. For propolis from Serbia, it has been determined by ultra HPLC-MS that *Populus nigra* is the plant used by bees [72]. Furthermore, using UPLC coupled to photodiode array detection and MS (UPLC-PDA-MS), Okinczyc et al. [73] demonstrated that, for propolis from Poland, Germany and Canada, *P. nigra* and *P. tremula* are the botanical sources. For propolis from different regions of Europe, such as Latvia, Russia, Finland, Poland, Ukraine, Slovakia, and France, it has been determined by GC-MS that *P. nigra*, *P. tremula,* and *Betula pubescens* are the plants used for production [74]. In addition, TLC and HPLC have been used to show that the *Zuccagnia punctata* plant is the botanical source of propolis produced in Argentina [75]. For propolis produced in Cuba, HPLC-MS has been used to show that *D. ecastophyllum* is the botanical source [76].

For propolis from the same country but from distinct regions, the plants used by bees can vary, as found by various authors who, through HPLC, identified that red propolis produced in the states of Sergipe [77,78], Alagoas [78,79], Bahia [78,80], Pernambuco, and Paraíba [78] in northeastern Brazil has *D. ecastophyllum* as its botanical source. The same plant was described by Silva et al. [81], who used GC-MS and HPLC to show that this same type of propolis is produced in the state of Alagoas in Brazil. Similarly, in 2020, Ccana-Ccapatinta et al. [82] used HPLC to show that the plants used by bees to produce this same type of propolis in the state of Bahia in Brazil are *Symphonia globulifera* and *D. ecastophyllum*. Other investigations have used distinct chemical analysis methods, such as HPLC [83,84], GC-MS [84,85], HPLC-MS, and GC with flame ionization detector (GC-FID) [84], to show that the botanical source of green propolis from the state of Minas Gerais in southeastern Brazil is *B. dracunculifolia*. HPLC-MS/MS has been used to show that, for this type of propolis produced in the state of Rio Grande in northeastern Brazil, *Mimosa tenuiflora* is the plant used by bees [86]. Adelmann et al. [87] used GC-MS to verify that *Populus deltoides* is indeed the botanical source for propolis produced in the mid-south region of the state of Paraná in southern Brazil.

Some studies have even attempted to classify propolis from Brazil based on its chemical composition and botanical source. Park et al. [88,89] described 12 different groups of Brazilian propolis, of which six groups are found in northeastern Brazil, five groups in the south and one group in the southeastern part of the country. In addition, in other investigations by these same authors, it was verified by high-performance thin-layer chromatography (HPTLC), HPLC, and GC-MS that for propolis from group 6 (northeastern part of the country), group 3 (south), and group 12 (southeast), the botanical sources are *Hyptis divaricata*, *P. nigra* [89], and *B. dracunculifolia* [89,90], respectively.

It is noteworthy that there is more literature describing the plants used by *A. mellifera* bees to produce propolis from Brazil compared with what has been reported for propolis from other regions of the world, but this should be studied in the other types of propolis, since the botanical diversity is very wide and identifying the floral sources, which are one of the factors that determine the chemical composition of propolis, would enrich the related body of literature.

In addition, there is another type of propolis, called geopropolis, which is produced by stingless bees. There is even less literature on its botanical source compared with the reports on *A. mellifera* and the propolis it produces. In this regard, Georgieva et al. [91] reported through GC-MS that *M. indica*, *Cratoxylum cochinchinense*, and *Dracaena cochinchinensis* plants are used for Vietnamese propolis produced by the stingless bee *Lisotrigona cacciae*. In addition, for propolis from this same country but produced by *Trigona minor*, NMR has been used to show that *M. indica* is the botanical source [92]. It has also been identified by HPLC that this same plant is used by *Tetragonula sapiens* for the production of Indonesian propolis [93]. In other investigations, it has been verified by GC-MS and NMR [94] or HPLC/ESI-MS [95] that *Tetragonula laeviceps* and *Tetragonula pagdeni* use the *Garcinia mangostana* plant to produce propolis in Thailand. There are also reports on *Tetragonula carbobaria* that have used HPLC, HPLC/ESI-MS, and NMR to show that these bees use the *Corymbia torelliana* plant to produce propolis in Australia [96]. Ferreira et al. [97] showed, using HPLC-DAD-MS/MS, that *Mimosa tenuiflora* is the botanical source used by *Scaptotrigona postica* for the production of propolis in the state of Rio Grande in Brazil.

It might seem that there is a large body of literature on the plants used by bees to produce propolis, but there are relatively few compared with all of the publications that exist regarding investigations carried out on the various activities of this bee product. In addition, for propolis from some geographical regions, such as Mexico, there have been almost no comparative studies on propolis and its botanical source. This is a very broad field of research considering that there is great floral diversity in Mexico. Researchers could try to classify propolis based on the floral origin, as has been done in other parts of the world, such as Brazil [88,89].

In contrast to the lack of information regarding the species of bee or the lack of investigations on the botanical source of propolis in certain countries, the type of solvent with which this bee product is extracted is often reported in the publications as ethanolic, methanolic, hexanic, or aqueous extracts, among others. In all of the articles reviewed here, the type of solvent utilized by the authors to obtain propolis extracts is mentioned: ethanol [39,44,45,46], methanol [40,41,42] or a combination of ethanol/water [43].

The review carried out by Suran et al. [98], describes, in a very detailed way, the research that has been carried out in recent years regarding the types of solvents used for the extraction of different types of propolis, both in a traditional way and with new technologies. For a few decades, methanol has been patented as an extraction method for propolis by maceration [99]. However, the use of 70% ethanol as the main solvent to extract propolis by maceration has been widely described [100]. The use of water for the extraction of propolis by maceration has also been mentioned [101]. Since it is a more polar solvent than ethanol, it has the ability to extract more polar compounds found in propolis [98]. Another solvent with a polar nature that has been shown to be highly effective for extracting a high percentage of polyphenols in propolis by maceration is 1,2-propylene glycol. It is widely used as a diluent or excipient in various pharmaceutical products, as it is safe and relatively non-toxic [102]. Similarly, among the non-polar solvents that can be used for the extraction of propolis, there are vegetable oils, since they mainly extract non-polar compounds [98]. It has been mentioned that, in propolis, they extract phenols [103]. In addition, some authors have mentioned that the methods of extraction of propolis by ultrasound and with a combination of ethanol and water have levels of efficiency similar to those of traditional maceration methods [104], or they may be even more efficient in terms of the extraction of active compounds in propolis [105]. Some innovative extraction methods include the use of natural deep eutectic solvents, such as the combination of lactic acid with water or choline chloride with 1,2-propylene glycol, which have been reported to be very efficient alternatives for propolis extraction. They are comparable to obtaining propolis with 70% ethanol and have the advantage of being safe and having low toxicity [106].

In some of the studies reviewed here, propolis extracts were partially purified with solvents of different polarity, such as hexane, dichloromethane, methanol, or ethyl acetate, to obtain various partitions [39,41,42]. Similarly, from some partitions of the extract, several fractions have been isolated by distinct column chromatographies [39,40,42], since propolis has a complex chemical composition. These studies sought to test the activity of some partitions or fractions that contained compounds similar in nature to the solvents used to extract them.

In this regard, new methods have recently been used for the extraction of different groups of compounds found in propolis, as reported by various authors [107,108] who described that the combination of different essential oils with supercritical carbon dioxide improves the extraction of flavonoids from propolis, since this technique can dissolve non-polar to slightly polar compounds. Another innovative and efficient method for the extraction of phenols and flavonoids in propolis is the vacuum resistive heating extraction technique, which consists of an ohmic vacuum heating process in combination with water and, later, 70% ethanol. This new technology improves the extraction efficiency, since the authors reported that they obtained approximately twice the concentration of phenolic compounds and about five times more flavonoids than with the traditional maceration technique [109]. Research on these new extraction technologies has been directed toward improving the ability to obtain different groups of compounds found in propolis. The type of extraction and solvent used in each investigation should be justified in order to obtain the best results and not rule out any natural product for not presenting activity when the best extraction solvent was not used.

In addition, it is important to consider the new alternative methods and solvents to obtain propolis extract that are currently being investigated, since if propolis, or some of its bioactive compounds, could be used as candidates for the treatment of gastric cancer, it is necessary to use extraction solvents that are safe, non-toxic, and can be used in the pharmaceutical industry.

Regarding the description of the chemical composition of propolis, there have been enormous advances since the beginning of the 21st century, mainly in the last decade [17,19,23,50,51,55,61,91,110,111]. In all of these investigations, significant effort has been made to identify more compounds in propolis through distinct analysis techniques, so there is a broad overview of the different groups of compounds that constitute the various types of propolis found around the world. Due to the exhaustive investigations carried out, more than 600 different chemical compounds have been identified [24] that are part of the complex chemical composition of propolis existing around the world. That is why we can state that propolis produced by *A. mellifera* contains a wide variety of chemical groups such as flavonoids (flavans, isoflavans, flavonols, flavanonols, flavanones, flavones, isoflavones, isodihydroflavones, chalcones, dihydrochalcones, and neoflavonoids), terpenes (monoterpenes, sesquiterpenes, diterpenes, triterpenes), phenols (phenylpropanoids, chlorogenic acids, stilbenes, lignans), phenolic acids, and other organic compounds such as fatty acids, aliphatic hydrocarbons, aldehydes, aliphatic acids and their esters, aliphatic fatty acids, aromatic acids and their esters, acetophenones, amino acids, sugars, and mineral elements [17,19,23,61,110].

In addition, the propolis produced by stingless bees has been described as having a more complex chemical composition than the propolis produced by *A. mellifera* [50]. In this type of propolis, the groups of compounds mentioned above and some others, such as alkylresorcinols, xanthones, anacardic acids, and alkaloids, have also been identified [50,51,55,91,111]. In fact, alkaloids have never been described in propolis produced by *A. mellifera*, but they have been found in some types of Brazilian propolis [112,113,114] produced by different species of stingless bee. The presence of alkaloids could explain, in part, the activity against gastric cancer of propolis produced by stingless bees from Thailand [39], Indonesia [41,42], and the Philippines [46] reviewed here, since it is known that these chemical compounds present cytotoxic activity through different mechanisms [115,116,117]. This is why it is essential that in the investigations carried out with propolis, the components that constitute its chemical composition are identified.

On the other hand, the chemical characterization of propolis samples takes a relevant point of interest from a toxicological view, because it is known that some chemicals such as pesticides and heavy metals produced as a consequence of human industrial and agricultural activities can contaminate bee products such as propolis [118,119]; in this sense, it has been reported that some samples of propolis from Poland were significantly contaminated with Pb and Mg, although a few concentrations also presented some traces of other heavy metals [118]. Additionally, Hodel et al. [119] reported the presence of As, Cd, and Pb in 19 representative samples of raw brown, green, red, and yellow Brazilian propolis, of which seven propolis samples exceeded the limits established by Brazilian regulations. This is relevant because Brazil is a major producer and exporter of propolis worldwide, although the presence of trace contaminants in this bee-derived natural product is limited [119,120].

However, de Oliveira Orsi et al. [121] reported that there was a reduction in the transfer rate of Ni, Cr, Hg, Cd, Pb, and Sn from raw Brazilian propolis samples to ethanolic extracts made with the propolis evaluated by this research group, allowing the authors to conclude that this decrease in the presence of toxic metals could make this bee-derivate product safe for use and consumption. This should be considered in the manufacturing process for commercially processed propolis to reduce the presence of contaminants in these manufactured products, since, as the study of González-Martín et al. [122] showed, there were heavy metals (Cr, Ni, Cu, Zn, and Pb) as well as pesticide residues (fungicides, herbicides, and acaricides) present in 31 commercial samples of propolis that included capsules tablets, tinctures, candies, and syrups from diverse countries (Spain, Portugal, Belgium, England, USA, and Chile).

Heavy metal contamination in propolis has relevance because it is known that these toxic metals can contribute to the incidence and mortality of gastric cancer; these elements act at different levels, increasing risk factors that can trigger carcinogenic processes in the stomach, including the disruption of gastric mucosal barrier integrity, the induction, directly or indirectly, of ROS generation and damage, both mucosal and at the DNA level, as well as the capacity to inhibit the DNA damage repair process, thus potentially leading to the induction of gene abnormalities. Finally, these contaminants are known to induce the proinflammatory process and microRNAs that can promote the tumorigenic process in the stomach [123,124].

Pesticide contamination also should have important relevance when chemical characterization of diverse propolis samples is carried out because these compounds are known as initiators of the carcinogenic process, although the association of pesticides with gastric cancer is little studied and unclear today. Therefore, most research should be carried out in this context [125,126,127,128]. However, contradictory reports on the presence of pesticides in propolis samples exist, such as the report by Chen et al. [129], who detected 4,4′-DDE, β-HCH, δ-HCH, and heptachlor in Chinese propolis, whereas in contrast, Zhou et al. [130] did not detect the presence of any pesticides in diverse Chinese propolis samples. In this line, Orsi et al. [120] investigated the presence of pesticides including organochlorines, organophosphates, pyrethroids, carbamates, herbicides, fungicides, and acaricides in 50 samples of Brazilian propolis by gas chromatography analyses but did not detect pesticide residues in any propolis sample.

There is no doubt that the chemical analysis of propolis is fundamental to identify its botanical origin, but the use of this technique can also show us the enormous variety of compounds that are components of the different types of propolis found around the world and that, undoubtedly, the vast majority of them have various properties that can benefit human health.

Regarding the types of evaluations carried out on the different types of propolis in the eight studies analyzed here, it is notable that most focused on in vitro tests with different gastric cancer cell lines (Table 1). Most of these cell lines were of Japanese origin, and this is due to the hard work carried out by Japanese researchers in response to the high incidence and mortality rate of gastric cancer in that country [131]. In seven of the eight studies analyzed in this systematic review [39,40,41,42,43,44,46], cell lines representing the histological types of gastric cancer were used. They were cancer cell lines derived from poorly (e.g., KATO-III, SGC-7901, NUGC-4, and MKN-45) [132,133,134], moderately (e.g., AGS cells) [134], or well-differentiated cells (e.g., NCI-N87 and MKN-74 cells) [135] from human gastric carcinomas. Although the KATO-III cell line was the most frequently used in the works already mentioned, the most interesting finding arose when analyzing the sets of cells, since we observed that propolis extracts from the different countries had activity against different gastric cancer cell lines. This is of great relevance, since each cell line has different morphological, genetic, and metabolic characteristics. Thus, propolis could have more than one therapeutic target in the elimination of tumor cells from the stomach. The effect on NCI-N87 cells was remarkable, since these cells have distinctive characteristics. They can grow in a colonial pattern, have confluence and form a monolayer, have high expression of human gastric lipase, fundic-type pepsinogen-5, MUC6, the zonula adherents marker E-cadherin, and the zonula occludens marker ZO-1, and respond to *H. pylori* infection in a manner more like primary gastric epithelial cell preparations [135,136]. It is worth mentioning that most gastric cancer cell lines lack such characteristics and lack important epithelial and/or glandular properties. Therefore, NCI-N87 cells are a study model that more closely resembles the tumor environment that develops in patients. In this way, the effects of propolis on different alterations related to the environment and tumor microenvironment in gastric cancer could be studied.

When analyzing the chemical compositions of propolis used in the articles included here (Table 1), we identified representative compounds found frequently in many propolis types. One example is CAPE, which has been widely reported in many types of propolis, such as those of Brazilian origin. It has been reported that this compound can act on NCI-N87 cell matrix molecules, negatively regulating protein levels of matrix metalloproteinase-9 (MMP-9) and vascular endothelial growth factor (VEGF), while increasing the levels of endostatin and thrombospondin-1 [137]. This is very interesting because these molecules are closely related to angiogenesis and tumor metastasis. In addition, CAPE inhibits *H. pylori*-induced NF-κβ DNA-binding activity and prevents IkB-α degradation in gastric cancer AGS cells [138]. Even in in vivo models of Helicobacter pylori-induced gastritis in Mongolian gerbils, CAPE can prevent NF-κβ activation and decrease COX-2, iNOS, IL-1β, and TNF-α mRNA expression [139]. Therefore, CAPE and propolis from different regions are very promising options in the study of gastric cancer and various alterations related to this disease.

Similarly, quercetin is one of the most frequently identified compounds in the different types of propolis. Different activities have been proposed for this compound through in silico studies and data mining. We highlight the work carried out by Yang et al. (2020), who identified six hub targets through a docking analysis: AKT1, EGFR, SRC, IGF1R, PTK2, and KDR [140]. These genes are directly related to tumor progression, a negative prognosis, and poor survival. This evidence is complemented by that reported by Shang et al. (2017), who showed that quercetin can induce apoptosis in gastric cancer AGS cells by increasing the concentrations of proapoptotic proteins such as Bad, Bax, and Bid while decreasing the levels of Mcl-1, Bcl-2, and Bcl-x, which are antiapoptotic proteins. Additionally, quercetin can increase the gene expression of tumor necrosis factor receptor superfamily, member 10d, decoy with truncated death domain (TNFRSF10D) and tumor protein p53 inducible nuclear protein 1 (TP53INP1) but decrease the gene expression of VEGF and cyclin-dependent kinase 10 (CDK10) [141]. Although the evidence provides a novel approach to reveal some of the possible therapeutic mechanisms of propolis and its components against gastric cancer, much of this evidence has been collected in vitro, so it is still necessary to carry out more in vivo work to obtain more information on diverse antitumor mechanisms that will facilitate future clinical applications.

In vitro studies have the advantage of being carried out under controlled conditions and with few variable parameters. In addition to providing information regarding whether or not propolis has anticancer activity, various data on the damage or elimination of the cell under study were provided in six of the eight articles reviewed here. However, one of the limitations of in vitro studies is that it is not possible to investigate the entire situation of the tumor microenvironment in an organism or to consider its metabolism since the activity of propolis is only being tested on a single cell type. That is why the advantage of in vivo studies lies in the fact that the activity of propolis on an organism can be determined, since these models provide us with great insight into the behavior of gastric cancer with a method of development similar to that in humans [142]. Furthermore, chronic *H. pylori* infection is known to promote the development of gastric cancer in humans [143]. Therefore, it has been reported that a model that presents greater similarity to the development of gastric cancer in humans is the use of MNNG together with the implementation of *H. pylori* infection in rodents [143,144].

However, one of the most frequently used in vivo models that has been employed for several decades to study gastric cancer is the MNNG-induced rat model [142,144,145], in which rare mutations in the p53, Ki-ras, and β-catenin genes have been reported [145]. Similarly, different genes involved in extracellular matrix remodeling, such as MMP3 and collagen types 1, 3 and 5, are up-regulated, and some genes associated with aldehyde dehydrogenase and aldose A (hydrocarbon metabolism), ion transporters (gastric juice), mucin 5 (mucus production), and gastrin and somatostatin (gastric hormones) are down-regulated in this model of gastric cancer. This gene expression profile found in stomach carcinomas from these animals has been shown to share many features with human stomach carcinomas [142]. In this regard, it has also been described that some genes associated with motility, adhesion and the cell cycle, matrix remodeling, and growth factors showed alterations in human stomach carcinomas [146,147].

In one study [45] reviewed here, one of the models used to evaluate the effect of Iranian propolis in vivo was the MNNG-induced rat gastric cancer tumor, which provided more information compared with the other six studies that only performed in vitro research. Although there has only been one report on propolis, it is necessary to specify that some work has been carried out with flavonoids, which are usually found in this bee product. For example, it has been reported that oral administration of naringenin to rats subjected to MNNG-induced gastric cancer tumors has a gastroprotective effect against the oxidative damage that characterizes the model [148]. In addition, it restored levels of cytochrome P450, cytochrome b5, NADPH cytochrome c reductase, LPO, SOD, CAT, GSH, and Vitamins C and E in both the stomach and the liver. It even prevented a loss of body weight, decreased the volume of tumors, and reduced the number of animals with gastric cancer in relation to the control group [149,150]. Considering this evidence, it is justifiable to carry out more in vivo studies with propolis containing naringenin, since the presence of other flavonoids or other phenolic compounds, such as CAPE, could lead to synergistic effects and have various anti-tumor effects.

In addition, regarding the pro-apoptotic effect of quercetin, it has been shown that this flavonoid acts through the induction of the expression of Bax and caspase-3 proteins and, in contrast, by reducing the level of Bcl2 in BGC-823 gastric carcinoma cells [151]. Additionally, kaempferol (another flavonoid reported to be part of the chemical composition of Iranian propolis) has also been reported to have pro-apoptotic effects on gastric cancer cell lines (MKN28 and SGC7901) with mechanisms similar to those reported for the quercetin pro-apoptotic pathways [152]. This is in line with the reported pro-apoptotic effect of Iranian propolis on an in vivo gastric cancer model. This is relevant because the low bioavailability of the flavonoids is well known and this directly affects the application of natural products in preclinical and clinical trials [153,154]. Therefore, the different flavonoids present in the chemical compositions of propolis samples have the advantage of displaying their anticancer effects in in vivo models, such as the MNNG-induced model.

Moreover, other anti-cancer gastric effects displayed by quercetin could also play essential roles in the activity of Iranian propolis. Among these is the regulation of the expression of cytochrome P450 enzyme, which can inhibit the activation of procarcinogens as well as enhancing the effect of quercetin on the DNA repair process. Together, these effects impact the up-regulation of phase II conjugated enzymes and could eliminate carcinogenic products and enhance or promote the mechanism involved in their elimination from the organism [155,156]. Furthermore, the effects impact the cell cycle arrest displayed by kaempferol. In addition, this same flavonoid influences the induction of autophagy through the activation of IRE1-JNK-CHOP signaling from the cytosol to the nucleus and G9a inhibition (HDAC/G9a axis) in gastric cancer cells [157]. Therefore, the possible synergic effects of the flavonoids present in Iranian propolis and the other anticancer mechanisms displayed by these compounds should be investigated in future work [45].

Additionally, it is important to state that these flavonoids are not the only compounds present in propolis. This bee-derived natural product also contains other secondary metabolites, such as terpenoids and phenolic acids, which may be closely associated with the anti-gastric cancer effect of propolis [46]. In work carried out with propolis from the Philippines, the authors focused on both the in vitro and in vivo anticancer effects of their propolis sample. Moreover, a crucial point in their research was the source of their propolis, because it was obtained from the stingless bee *Tetragonula biroi* Friese (syn *Trigona biroi*) [46]. This is in contrast to Iranian propolis, which is made by the most common bee, *A. mellifera* [45]. Apart from differences in the chemical compositions of these two propolis samples, propolis from the Philippines displayed in vivo anticancer effects mainly through cell cycle arrest, whereas the induction of apoptosis to reduce gastric pyloric tumors was apparently less important in this Pacific-type propolis. This is in contrast to the pro-apoptotic effects displayed for this propolis sample in diverse gastric cancer cell lines [46].

Regarding the former, it may be not only necessary to investigate the botanical source of propolis, but also important to consider the bee species that makes the propolis sample. This is because research on each propolis sample, both in vitro and in vivo, helps to form a better understanding of the possible mechanism involved in the anti-gastric cancer properties of these natural products. Finally, it is evident that, although in vitro studies of the effects of propolis samples against gastric cancer cell lines are important for identification of the anticancer properties of propolis as well as to determine the possible mechanisms involved, is necessary to carry out this same research using in vivo models, as these can supplement the information obtained regarding possible applications of propolis to develop alternative treatments for gastric cancer.

## 5. Conclusions

The present study analyzed the information available to date on propolis and its effects on gastric cancer. Seven of the eight articles used in vitro models, and only two of them included experimental animal designs. Although the solvents, extracts, and components of propolis differed among the studies analyzed, the authors demonstrated that propolis has the ability to inhibit the cell viability in all cell lineages, mainly by regulating the proteins responsible for cell cycle progression and through the induction of an apoptosis mechanism. In the articles that used in vivo models, the researchers also described the regression of tumors at the macroscopic and histological levels and correlated the activity of propolis with both cell cycle arrest and the induction of apoptosis. These were deemed to be the principal mechanisms by which this bee-derived natural product displays its anticancer properties. These are in addition to those shown in general by other components in the chemical compositions of propolis samples. However, the relationships of these secondary metabolites with the propolis anticancer properties were not clear in these works. We propose a relationship between the effects of propolis and the chemical compositions reported in the works included in this study. We found a strong association between compounds of phenolic origin (mainly flavonoids) and various antitumor effects on cell lines and in in vivo models of gastric cancer. Therefore, we believe that the chemical compositions of the different types of propolis may be responsible for protecting against gastric cancer.

The evidence is very interesting, but more research is required to understand with certainty how propolis can contribute to the treatment of gastric cancer. We need to fundamentally understand which of its components can effectively regulate cell cycle proteins to inhibit proliferation and stop the progression of the neoplasia in patients. It is also necessary to carry out more preclinical trials with known propolis types that have already been identified as having anticancer properties. In addition, new and known types of propolis with different origins that could be prominent agents in the development of alternative strategies for the treatment of gastric cancer should be researched through clinical assays.

To address this challenge, we consider that it is necessary for future studies to describe the chemical composition of the propolis used. This will make it easier to understand and explain the antitumor effects of propolis as well as to identify the responsible compounds and determine the precise doses required for adequate and safe consumption. Therefore, we consider that this systematic review is sufficient to encourage further research on the administration of propolis as a therapeutic (and even prophylactic) regimen to help patients before surgery.

## Figures and Tables

**Figure 1 foods-12-00415-f001:**
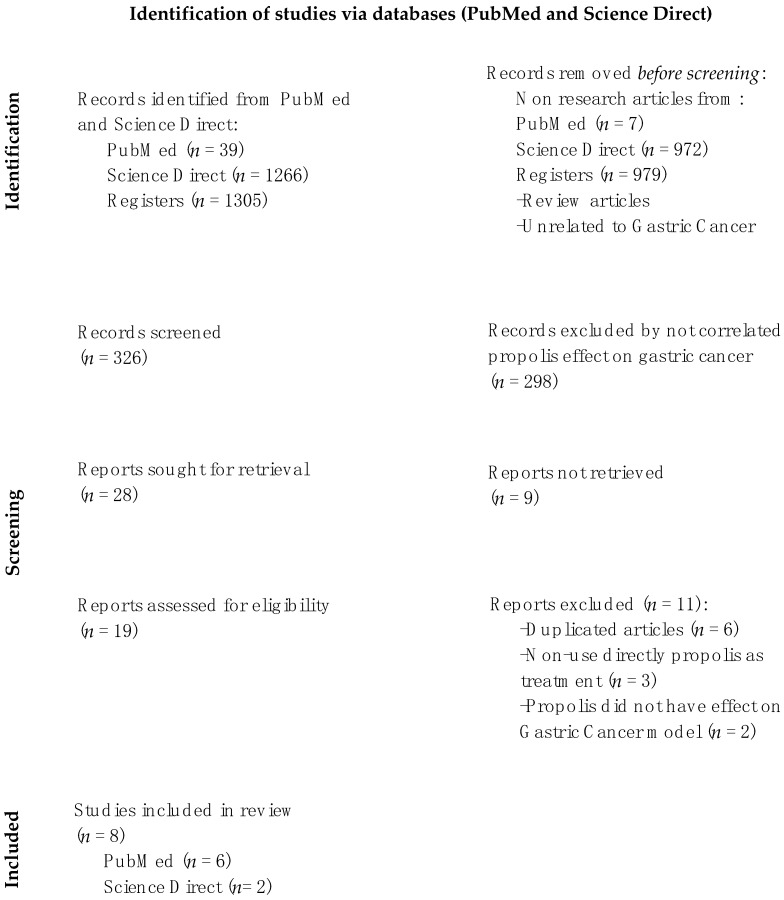
PRISMA flow diagram of literature selection process used in this review, applied to both the PubMed and Science Direct databases.

**Figure 2 foods-12-00415-f002:**
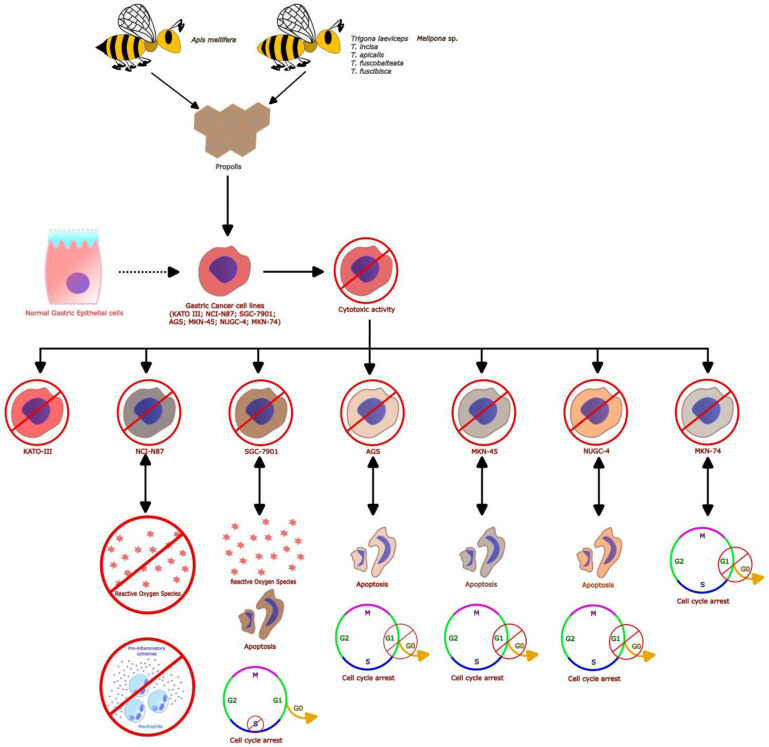
Schematic representation of the principal gastric cancer cell lines tested to determine the anticancer effects of diverse propolis samples.

**Figure 3 foods-12-00415-f003:**
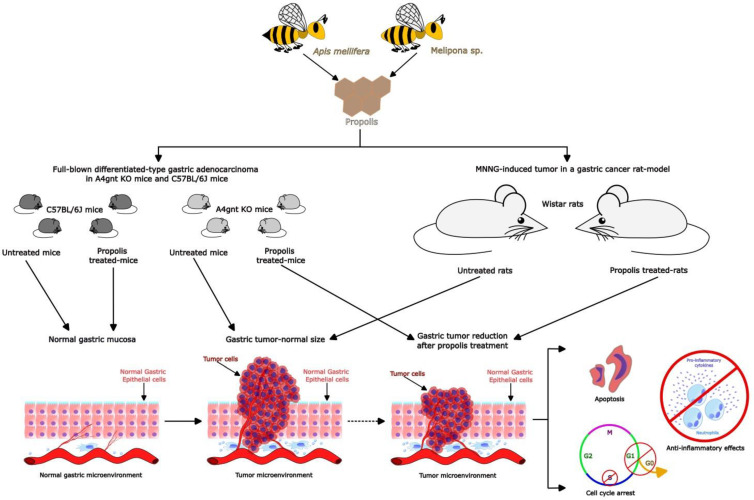
Schematic representation of the in vivo gastric cancer models used to test diverse propolis samples and their possible mechanisms to display their anticancer effects.

**Table 1 foods-12-00415-t001:** Reports of in vitro and in vivo effects of propolis from studies on gastric cancer performed in different countries.

Propolis (Country)	Components Identified	Experimental Model and Protocol	Results Obtained	Ref.
Thailand (stingless bee)	-N.I.	Model: Gastric carcinoma KATO-III (ATTC No. HTB 103) cell line. Protocol: In vitro cytotoxic activity was assessed by the MTT method. Liver (CH-liver) and fibroblasts (HS-27) were used for a comparison.	-Ethanol extract IC_50_ of 22.98 μg/mL. -Hexane extract obtained from the ethanol extract showed a IC_50_ of <20 μg/mL. -Partitions obtained from the hexane extract with IC_50_ of <20 μg/mL: Partitions of 10%, 30%, 50%, 70% and 100% dichloromethane, and 10% hexane. -Fractions obtained from 30% dichloromethane: F2 (IC_50_ of 18.07 μg/mL) and F3 (IC_50_ of 4.09 μg/mL); Fractions obtained from 100% dichloromethane: F3 (IC_50_ of 7.55 μg/mL) and F4 (IC_50_ of 8.31 μg/mL).	[39]
Thailand (*A. mellifera*)	-Cardanol -Cardol	Model: Gastric carcinoma KATO-III (ATCC No. HTB 103) cell line. Protocol: In vitro cytotoxic activity was assessed by the MTT method. A non-transformed human foreskin fibroblast cell line (Hs27, ATCC No. CRL 1634) was used for the comparison. Chemical analysis of the fractions by NMR and ESI-MS.	-Hexane extract IC_50_ of 42.5 ± 6.61 μg/mL. -Dichloromethane extract IC_50_ of 43.8 ± 6.5 μg/mL. -Fractions obtained from the hexane extract with cytotoxic activity: Fraction III IC_50_ of 13.69 ± 1.44 μg/mL. Fraction IV IC_50_ of 40.16 ± 2.66 μg/mL. Fraction V IC_50_ of 15.21 ± 2.13 μg/mL. -Compounds identified in fractions III and V obtained from hexane extract: Cardanol (fraction III) IC_50_ of 13.71 ± 1.42 μg/mL. Cardol (fraction V) IC_50_ of 8.78 ± 0.28 μg/mL.	[40]
Indonesia (stingless bee)	-N.I.	Model: Gastric carcinoma KATO-III (ATCC No. HTB 103) cell line. Protocol: In vitro cytotoxic activity was assessed by the MTT method.	-Extracts obtained from different bee species, with IC_50_ of 20 μg/mL: *T. incisa* methanol and ethyl acetate extracts *T. apicalis* hexane extract *T. fuscobalteata* methanol and ethyl acetate extracts *T. fuscibasis* methanol and ethyl acetate extracts -Kaempferol and apigenin (purchased compounds) showed IC_50_ of 10 μg/mL.	[41]
Indonesia (stingless bee)	-5-pentadecyl resorcinol (Cardol isomer) -Terpenoid-like pattern	Model: Gastric carcinoma KATO-III (ATCC No. HTB103) cell line. Protocol: In vitro cytotoxic activity was assessed by the MTT method. A normal skin fibroblast cell line (CCD-986 sk, ATCC No. CRL1947) was used for comparison. Chemical analysis of the fractions by NMR and ESI-MS.	-Ethyl acetate extract (partition) IC_50_ of 8.06 ± 0.08 μg/mL. -Fractions obtained from ethyl acetate extract with cytotoxic activity: F24 (IC_50_ of 9.35 ± 0 μg/mL). F26 (IC_50_ of 6.75 ± 1.15 μg/mL). F27 (IC_50_ of 6.61 ± 1.29 μg/mL). F36 (IC_50_ of 9.75 ± 0.19 μg/mL). F45 (IC_50_ of 6.06 ± 0.39 μg/mL); and 5-pentadecyl resorcinol as the main compound. F46 (IC_50_ of 8.25 ± 0.22 μg/mL); terpenoid-like pattern was identified in this impure fraction.	[42]
New Zealand	-CAPE -Pinobanksin -Pinobanksin-3-O-acetate -Pinocembrin -Chrysin -Galangin	Model: Human gastric cancer cells NCI-N87 (ATCC CRL-5822). Protocol: Production of different types of propolis-cyclodextrin complexes: CD1, CD2, CD3, CD4 and CD5. In vitro cytotoxic activity was assessed by the MTT method. Activated neutrophil anti-inflammatory assays. Lipid antioxidant assay. Positive control 5-FU tested at 15 ng/mL. Compounds reported in propolis were given by the manufacturer of this sample.	-Cytotoxic activity: Propolis complexes had moderate cytotoxic activity since CD3 inhibited NCI-N87 cells by 32.7%, CD4 by 24.6%, and CD5 by 21.8% at 200 μg/mL. Pinocembrin had 72.5% cytotoxic activity at 200 μg/mL. -Anti-inflammatory activity: At 50 μg/mL, New Zealand propolis (alone) inhibited TNF-α by 85% ± 1, CD1 by 93% ± 1, and CD2 by 97% ± 1. At 200 μg/mL, all three samples inhibited this cytokine by 100%. -Lipid antioxidant activity: The five propolis complexes and CAPE (also in the γ-CD complex) had moderate antioxidant activity. CAPE (alone) showed strong antioxidant activity.	[43]
China	-Caffeic acid -p-Coumaric acid -Ferulic acid -Isoferulic acid -3,4-Dimethoxycinnamic acid -Pinobanksin -Naringenin -Quercetin -Kaempferol -Apigenin -Pinocembrin -Benzyl caffeate -3-O-Acetyl pinobanksin -Chrysin -CAPE -Galangin -Benzyl p-coumarate	Model: Cell line SGC-7901 Protocol: Cell viability was measured through the CCK-8 assay, and the morphological changes were examined with a microscopical technique. Apoptosis, cell cycle arrest, ROS generation, and changes in the mitochondrial membrane permeability were detected by the Annexin V-FITC/PI, PI, DCFH-DA, and JC-1 flow cytometry protocols, respectively. Cytochrome C, Cleaved PARP, tubulin CDK2, CDC2, E2F1, P-Rb, Cyclin A2, Cyclin E, Bcl-2, Cleaved Caspase-3, Cleaved Caspase-8, P-53, Bid, Bax, and Cleaved Caspase-9 were analyzed by Western blot assay. Chemical analysis of propolis by HPLC.	-Ethanolic propolis extract displayed an IC_50_ of 66.64 µg/mL in SGC-7901 cells. Moreover, it induced shrinking, loosening, and a decrease in the number of cells in plates analyzed by microscopy. -Propolis induced ROS generation and a loss in mitochondrial membrane permeability in SGC-7901 cells. -Apoptosis induced in SGC-7901 cells by propolis was related to the upregulation of the proteins Bax and Bid, the down-regulation of Bcl-2, and the activation of Cleaved Caspase-8, Cleaved Caspase-9, Cleaved Caspase-3, Cleaved PARP, and P-53. -S-phase arrest induced by propolis in SGC-7901 cells was associated with the dose- and time-dependent up-regulation of P-Rb, CDC2, CDK2, Cyclin E, Cyclin A2, and E2F1 expression.	[44]
Iran	In both propolis: -Caffeic acid -Caffeic acid isoprenyl ester -Ferrulic acid -Isoferrulic acid -P-coumaric acid -Quercetin -Quercetin-3 methyl ether -Quercetin-7 methyl ether -Kaempferol -Pinobanksin -Pinobanksin 5,7-dimethyl ether -Pinobanksin 3 methyl ether -Pinobanksin -3-O-acetate -Pinobanksin-3-O-proprionate -Pinobanksin-3-O-butyrate -Pinobanksin-3-O-pentanoate -Luteolin-5-methyl ether	Model: MNNG-induced tumor in a gastric cancer model Protocol: 55 Wistar rats were divided in 3 experimental groups: Control (*n* = 15), Taleghan propolis (*n* = 20) and Hamadan propolis (*n* = 20). All groups were treated with 100 μg/mL of MNNG in drinking water *ad libitum* for 34 weeks. Propolis-treated groups (ethanolic extract [500 mg/mL]) began propolis consumption prior to two weeks of MNNG administration. Observations the tumor type and presence of metastases, incidence, number, and size of tumors were made. A histological analysis was performed by hematoxylin-eosin (H&E) staining. Additionally, β-catenin, Bax, and Bcl2 antibodies were determined by immunohistochemistry analysis.	-The incidence and number of tumors were significantly decreased by propolis with respect to the control group. -The expression of the nuclear/cytoplasm ratio, epithelial stratification, nuclear dispolarity, structural abnormality, and b-catechin and Bcl-2 protein were decreased in propolis groups with respect to the control group. -Propolis groups showed increased expression of the Bax protein with respect to the control group. -The evidence shows that Iranian propolis exerts inhibitory effects on cell proliferation and apoptosis induction against MNNG-initiated gastric cancer.	[45]
Philippine (stingless bee)	-Guaiol -Tibolone -Andrographolide -Gallic acid -β-Eudesmol -Danthron -Ginkgolide-B -Colchicine -Cinnamic acid -Protocatechuic acid -Ginkgolic acid -Rhodoxanthin -Pterostilbene -Rosmanol -Butylated hydroxytoluene	Model: In vitro model: Human gastric cancer cells lines (AGS, MKN-45, NUGC-4, MKN-74). In vivo model: Full-blown differentiated-type gastric adenocarcinoma in A4gnt KO mice and C57BL/6J mice. Protocol: In vitro model: In vitro cytotoxic activity was assessed by the MTT method from 1 μg/mL to 1000 μg/mL at 24, 48, and 72 h. Cisplatin was used as a positive control under same conditions. The cell cycle arrest assay was performed by flow cytometry with only AGS cells cultured at a concentration of 188 μg/mL of propolis (this represents the IC_50_ of the sample at 48 h). Induction of apoptosis of propolis samples on four gastric cancer cell lines was determined by the TUNEL assay; moreover, this parameter also was evaluated on histological sections obtained by in vivo assays. In vitro model: 14 C57BL/6J mice and 20 A4gnt KO mice were divided into four experimental groups: C57BL/6J + distilled water (*n* = 7), C57BL/6J + propolis (*n* = 7), A4gnt KO mice + distilled water (*n* = 10) and A4gnt KO mice + propolis (*n* = 10). Groups treated with 100 mg/kg for 30 days. Macroscopical gross gastric mucosal elevation and the histological thickness of the gastric mucosa were evaluated; moreover, an immunohistochemistry assay was performed to evaluate the expression of CD3, BrdU, and p21. Finally, gene expression in both mice and human gastric cancer samples was determined by qRT-PCR on the homogenized gastric cancer model and the four gastric cancer cell lines evaluated.	-Propolis showed the following values of IC_50_ (µg/mL) for different gastric cancer cell lines at 24, 48, and 72 h: AGS (650, 188, 39); MKN45 (1156, 386, 318); NUGC4 (580, 376, 315) and MKN74 (1259, 955, 925). -Propolis modulates the cell cycle and apoptosis through the regulation of gene expression in each gastric-cancer cell line, as follows: AGS: up-regulation of CDKN1A; demotion of the expression of CDK1 and CCND1; and increases in CDKN1A, CDKN1B, and TP53. With respect to apoptosis, the induction of Bax and Bad was observed. MKN-45: propensity for the up-regulation of CDKN1A; demotion of the expression of CDK1 and CCND1; and down-regulation of CDK2. With respect to apoptosis, down-regulation of the expression of Bcl-2L1 were observed. NUGC-4: up-regulation of CDKN1A; demotion of the expression of CDK1 and CCND1; and down-regulation of CDK2. With respect to apoptosis, the induction of Bax and Bad, and a down-regulation of the expression of Bcl-2 were observed. MKN-74: demotion in the expression of CDK1 and CCND1. In AGS cell line, propolis arrested at the G0/G1 phase in 66% of tested cells. It also increased the number of S-phase cells, and depleted cells at the G2/M and multi-nuclear phases. Additionally, propolis promoted DNA fragmentation in AGS, NUGC-4, and MKN-45 cells. -In vivo activity: In vivo treatment of A4gnt KO mice with propolis showed regression of the gross mucosal elevation at both macroscopical and histological levels and a reduction in CD3-positive T-lymphocytic cell infiltration. Treated C57BL/6J mice did not show any differences from untreated mice. C57BL/6J and A4gnt KO mice treated with propolis showed an increasing tendency for IL-10 transcription, modulation of cell-cycle protein-encoding genes, such as CDKN1A, increased the expression of CDKN1B, and a reduction in the expression of CDK1. Propolis supplementation by A4gnt KO mice decreased the number of actively dividing BrdU-positive S-phase cells.	[46]

N.I. = Not identified; NMR: Nuclear magnetic resonance; ESI-MS: Electrospray ionization-mass spectroscopy; CAPE: Caffeic acid phenethyl ester; 5-FU: 5-fluorouracil; ROS: Reactive oxygen species; HPLC: High-performance liquid chromatography; MNNG: N-methyl-N-nitro-N-nitrosoguanidine.

## Data Availability

The data are available from the corresponding author.

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
