# Peer review of "The Role of Propolis as a Natural Product with Potential Gastric Cancer Treatment Properties: A Systematic Review"

_foods, 2023, doi:10.3390/foods12020415_

Round 1

Reviewer 1 Report

The present article is a thorough review of a limited number of articles dedicated to the use of propolis in gastric cancer treatment, based on in vitro analysis with cell lines from gastric adenocarcinoma and in vivo murine models application. The Authors have used the PRISMA model, the review is of interest. The Table is well structured and informative. There are however, several points which need additional attention by the Authors.

1.       Major concern is as follows: One more item should be added to Exclusion criteria: chemical characterization of the propolis used in the experiments. The lack of chemical characterization makes the studies irreproducible and scientifically unsound. Based on this consideration, the references 39 and 42 should be removed from the review and the Table.

2.       The detailed discussion on potential propolis sources is not appropriate for such a review. The identified components in the studied samples are markers of specific sources. Cardols and cardanols come from Mangifera indica. The combination of caffeic acid esters, pinocembrin, pinobanksin acetate (and higher esters) are Populus markers.

3.       Page 18, line 542: “species of bees” should be changed to “subspecies of bees”, all the mentioned are subspecies of Apis mellifera.

Author Response

Reviewer 1

The present article is a thorough review of a limited number of articles dedicated to the use of propolis in gastric cancer treatment, based on in vitro analysis with cell lines from gastric adenocarcinoma and in vivo murine models application. The Authors have used the PRISMA model, the review is of interest. The Table is well structured and informative. There are however, several points which need additional attention by the Authors. 

  1. Major concern is as follows: One more item should be added to Exclusion criteria: chemical characterization of the propolis used in the experiments. The lack of chemical characterization makes the studies irreproducible and scientifically unsound. Based on this consideration, the references 39 and 42 should be removed from the review and the Table.

Reply

In relation to the suggestion of adding one more item to the exclusion criteria, we consider that said modification drastically affects the structure and proposal of this systematic review, since our intention is focused on analyzing the works that evaluate propolis against gastric cancer. For this reason, we consider that the methodology and exclusion criteria can remain as proposed. In relation to the chemical characterization, the suggestion is interesting, valuable and very appropriate with regard to the study of natural products such as propolis. However, it is precisely this limitation that we discuss and criticize, and we even propose that chemical analysis should be included in future work in order to perform any of the biological activities of propolis with greater precision. In our own previous works we have even emphasized the need to include a chemical analysis that allows knowing the compounds that could be responsible for the biological effects evaluated. Nevertheless, despite the fact that references 39 and 42 lack a chemical description, it is important to keep them in this systematic review since they are part of the few works that evaluate the effect of propolis and some aspect of gastric cancer.

  1. The detailed discussion on potential propolis sources is not appropriate for such a review. The identified components in the studied samples are markers of specific sources. Cardols and cardanols come from Mangifera indica. The combination of caffeic acid esters, pinocembrin, pinobanksin acetate (and higher esters) are Populus markers.

Reply

We agree with the suggestion and to avoid confusion about possible botanical sources of propolis without verification of a chemical analysis, the text from line 560 to line 565 and the text from line 576 to line 581 have been deleted. However, we consider that the text of lines 585 to 675, which refers to the botanical sources of propolis identified through different chemical analyses, is of great relevance in order to explain and understand the origin of the chemical composition of the different propolis. Therefore, we propose that this information be kept in the discussion. Additionally, as we mentioned from line 678 to 683: "In addition, for propolis from some geographical regions, such as Mexico, there have been almost no comparative studies on propolis and its botanical source. This is a very broad field of research considering "That there is great floral diversity in Mexico. Researchers could try to classify propolis based on the floral origin, as has been done in other parts of the world such as Brazil [90, 91]."

This information demonstrates the impact that the botanical source has on the chemical composition and therefore on the biological activities of propolis.

  1. Page 18, line 542: “species of bees” should be changed to “subspecies of bees”, all the mentioned are subspecies of Apis mellifera.

Reply

We agree with the suggestion. It was corrected in line 542 "species of bees" with "subspecies of bees".

Reviewer 2 Report

·       What about harmful chemical contaminants in propolis? I mean chemicals/xenobiotics of anthropogenic origin from the environment, such as PAH, pesticides, heavy metals and others.... Don't these chemicals negatively affect the human body and can they cause cancer? Please discuss this problem in relation to the topic.

·       MTT assay is not test for antiproliferation effect. It measures metabolic activity of cells. There are some other test for antiproliferation testing such as Brdu. In view of the above, the authors should consider using the term "antiproliferative effect" to refer to the MTT test and other tests that do not test it at all.

·       Table 1 is too elaborate and unfriendly to the reader. Should be limited to only the most important information, while much of the information in the table should be discussed in the text.

·       Reference list must be corrected because it is done carelessly.

Author Response

Reviewer 2

  • What about harmful chemical contaminants in propolis? I mean chemicals/xenobiotics of anthropogenic origin from the environment, such as PAH, pesticides, heavy metals and others.... Don't these chemicals negatively affect the human body and can they cause cancer? Please discuss this problem in relation to the topic.

Reply

The questioning is very valid and interesting, so we add the following in line 767: On the other hand, the chemical characterization of propolis samples takes a relevant point of interest from a toxicological view, because it is known that some chemicals such as pesticides and heavy metals produced as a consequence of human industrial and agricultural activities can contaminate the bee products such as propolis (Formicki et al., 2013; Hodel et al., 2020); in this sense, has been reported that some samples of Poland propolis were significantly contaminated with Pb and Mg, although in few concentration also presented some traces of other heavy metals (Formicki et al., 2013); besides, Hodel, K.V. et al., (Hodel et al., 2020) reports the presence of As, Cd, and Pb in 19 representative samples of raw brown, green, red, and yellow Brazilian propolis, of which seven propolis samples exceeded the limits established by the Brazilian regulation, this takes relevance because Brazil is a great producer and exporter of propolis in worldwide although the presence of trace contaminants in this bee-derived natural product is limited (Hodel et al., 2020; Orsi et al., 2012).

However, de Oliveira Orsi, R. et al., (de Oliveira Orsi et al., 2018) reported that there is a reduction in the transfer rate of Ni, Cr, Hg, Cd, Pb, and Sn from raw Brazilian propolis samples to ethanolic extracts made with the propolis evaluated by this research group, with which authors conclude that this decrease in the presence of toxic metals can make this bee-derivate product safe for their use and consumption; this should be considered in the manufacturing process of commercially processed propolis to reduce the presence of contaminants in these manufactured products, since, as the study of González-Martín, M. et al., (González-Martín et al., 2018) shows there are the presence of heavy metals (Cr, Ni, Cu, Zn, and Pb) as well as pesticides residues (fungicides, herbicides, and acaricides) in 31 commercial presentations of propolis that includes capsules tablets, tinctures, candies, and syrups from diverse countries (Spain, Portugal, Belgium, England, USA, and Chile).

Heavy metal contamination in propolis takes relevance because it is known that these toxic metals can contribute to the incidence and mortality of gastric cancer because these elements act at different levels increasing risk factors that can trigger carcinogenic process in the stomach, between these is found the disruption of gastric mucosal barrier integrity, the induction directly or indirectly of ROS generation and damage both mucosal and DNA level, as well as the capacity of heavy metals to inhibits DNA damage reparation process and therefore this can lead to the induction of gene abnormalities, and finally, these contaminants are known to induce the induction of proinflammatory process and microRNAs that can promote the tumorigenic process in the stomach (Wang et al., 2022; Yuan et al., 2016).

Pesticide contamination also should take an important relevance when chemical characterization of diverse propolis samples is carried out because these compounds are known as initiators of the carcinogenic process, although the association of pesticides with gastric cancer is little studied and unclear today, therefore, most research should be carried out in this sense (Lee et al., 2004; Mills & Yang, 2007; Santiago et al., 2021; Yildirim et al., 2014); although exist contradictory reports about their presence in punctual propolis samples such as the reported by Chen, F. et al., (Chen et al., 2009) whose detected 4,4´-DDE, b-HCH, d-HCH, and heptachlor in Chinese propolis; in counterpart, Zhou, J. et al., (ZHOU et al., 2005) did not detect any pesticide presence in diverse Chinese propolis samples. In this line, Orsi, R.d.O. et al., (Orsi et al., 2012) evaluates the presence of pesticides including organochlorines, organophosphates, pyrethroids, carbamates, herbicides, fungicides, and acaricides in 50 samples of Brazilian propolis by gas chromatography analyses; nevertheless, they did not detect pesticide residues in any propolis sample.

  • MTT assay is not test for antiproliferation effect. It measures metabolic activity of cells. There are some other test for antiproliferation testing such as Brdu. In view of the above, the authors should consider using the term "antiproliferative effect" to refer to the MTT test and other tests that do not test it at all.

Reply

We agree with the suggestion, we consider that the term "antiproliferative" is inappropriate, which is why it was replaced with the term "cytotoxic" in the lines: 41, 80, 168 (Table 1, it was replaced in references 39, 40, 41, 42, 43, 44, 45, and 46), 232, 283, 287, 296, 308, 317, 360, 369, 374, 378, 388, 393, 394, 396, 403, 406, 428, 430, and 435. Furthermore, the word "proliferation" was replaced by "viability" in lines: 388, 438, 450, and 921.

  • Table 1 is too elaborate and unfriendly to the reader. Should be limited to only the most important information, while much of the information in the table should be discussed in the text.

Reply

We understand the complexity and length of Table 1. However, we believe that the information summarized in Table 1 is necessary for the reader to understand all the aspects that we address in the article. In addition, the information summarized in Table 1, which is related to the articles of the systematic review, is mentioned in detail in the results section, from lines 159 to 520, first addressing the entire methodological part and later emphasizing the benefits of propolis for gastric cancer in both cell and animal models. Due to this, the authors consider that the table can remain with the original structure, however, we summarize the information a little more.

  • Reference list must be corrected because it is done carelessly.

Reply

The reference list was revised, corrected, and integrated using the "EndNote X8" program.
